# Antibody affinity and cross-variant neutralization of SARS-CoV-2 Omicron BA.1, BA.2 and BA.3 following third mRNA vaccination

Lorenza Bellusci[1,4], Gabrielle Grubbs [1,4], Fatema Tuz Zahra[1,4], David Forgacs[2], Hana Golding[1], Ted M. Ross[2,3] & Surender Khurana [1] ✉

There is limited knowledge on durability of neutralization capacity and antibody affinity maturation generated following two versus three doses of SARS-CoV-2 mRNA vaccines in naïve versus convalescent individuals (hybrid immunity) against the highly transmissible Omicron BA.1, BA.2 and BA.3 subvariants. Virus neutralization titers against the vaccine-homologous strain (WA1) and Omicron sublineages are measured in a pseudovirus neutralization assay (PsVNA). In addition, antibody binding and antibody affinity against spike proteins from WA1, BA.1, and BA.2 is determined using surface plasmon resonance (SPR). The convalescent individuals who after SARS-CoV-2 infection got vaccinated develop hybrid immunity that shows broader neutralization activity and cross-reactive antibody affinity maturation against the Omicron BA.1 and BA.2 after either second or third vaccination compared with naïve individuals. Neutralization activity correlates with antibody affinity against Omicron subvariants BA.1 and BA.2 spikes. Importantly, at four months post-third vaccination the neutralization activity and antibody affinity against the Omicron subvariants is maintained and trended higher for the individuals with hybrid immunity compared with naïve adults. These findings about hybrid immunity resulting in superior immune kinetics, breadth, and durable high affinity antibodies support the need for booster vaccinations to provide effective protection from emerging SARS-CoV-2 variants like the rapidly spreading Omicron subvariants.

The emergence of Omicron (B.1.1.529/BA.1) variant of SARS-CoV-2 in November 2021, and its rapid spread across the globe, resulted in designation of Omicron as a variant of concern (VOC)[1]. Viral spike protein of the Omicron variant contains higher number (>30) of mutations compared to other variants, raising concerns that Omicron is resistant to neutralizing antibodies generated following SARS-CoV-2

vaccination or infection, and a third vaccine dose was recommended to boost immunity.

Since then, several subvariants of Omicron emerged (BA.2 and BA.3), each with its own set of mutations (Supplementary Table S1)[2]. Thus far, the BA.2 demonstrates higher transmissibility compared with BA.1 and has reached dominance in several countries in Europe and

[1]Division of Viral Products, Center for Biologics Evaluation and Research (CBER), FDA, Silver Spring, MD 20871, USA. [2]Center for Vaccines and Immunology, University of Georgia, Athens, GA 30602, USA. [3]Department of Infectious Diseases, University of Georgia, Athens, GA 30602, USA. [4]These authors contributed equally: Lorenza Bellusci, Gabrielle Grubbs, Fatema Tuz Zahra. ✉e-mail: Surender.Khurana@fda.hhs.gov

Africa[3]. However, limited knowledge exists regarding vaccine-induced neutralizing and antibody affinity following two doses vs. three doses of mRNA vaccination in naïve vs. COVID-19 recovered individuals against SARS-CoV-2 Omicron BA.2 and BA.3 lineages and the durability of the vaccine-induced immunity. Therefore, it is critical to know the persistence of cross-reactive immunity generated following SARS-CoV-2 vaccination to neutralize the SARS-CoV-2 omicron subvariants BA.1, BA.2 and BA.3.

In this study, we evaluated the capacity and durability of neutralizing antibodies and antibody affinity induced following mRNA-based (Pfizer-BioNTech BNT162b2 or Moderna mRNA-1273) vaccination in naïve versus convalescent individuals (infection before vaccination) both at one-month after the second and third vaccinations and at 4 months post-3rd vaccination against vaccine-homologous WA1 and Omicron BA.1, BA.2 and BA.3 subvariants.

## Results

### Neutralizing antibodies following second and third vaccination in naïve and convalescent adults against SARS-CoV-2 WA1 and Omicron subvariants

In this study, we evaluated immune response following mRNA-based (Pfizer-BioNTech BNT162b2 or Moderna mRNA-1273) vaccination in a cohort of 81 adults: either naïve ($N = 50$) or SARS-CoV-2 convalescent (infection before vaccination; N = 31) individuals both at peak (1 month) immune response after the second and third vaccination, as well as at 4 months post-3rd vaccination, against vaccine-homologous WA1 and Omicron BA.1, BA.2 and BA.3 subvariants. There were no significant differences for age, sex, race, ethnicity, body mass index (BMI) or vaccine type between the two cohorts (Table 1 and Supplementary Table S2). The 31 convalescent individuals had SARS-CoV-2 infection between March – November 2020 (prior to emergence of Omicron). During that time, the predominant circulating strain in the US were the D614G strain and the Alpha variant prior to availability of mRNA vaccines. All subjects received the second dose of mRNA vaccines between January and February 2021 and a third vaccine dose between September-October 2021 either from Moderna (mRNA-1273) or from Pfizer (BNT162b2). The first two vaccine doses were homologous for each participant. The boosters were either homologous or

### Table 1 | Participant demographics and characteristics

| Participant demographics and vaccination type | Convalescent COVID-19 n(%) | Naïve n(%) | Significance p value |
|---|---|---|---|
| **Sex** | | | 0.7786 |
| Male | 10 (32) | 18 (35) | |
| Female | 21(68) | 33 (65) | |
| **Age mean (range) in years** | 46 (22–65) | 53 (30–81) | 0.1979 |
| **2nd dose** | | | 0.6683 |
| Pfizer | 27 (87) | 41 (82) | |
| Moderna | 4 (13) | 9 (18) | |
| **3rd dose** | | | 0.6013 |
| Pfizer | 27 (87) | 37 (74) | |
| Vaccine Type 3rd dose, Moderna | 4 (13) | 13 (26) | |
| **BMI mean** | 31 | 27 | 0.3529 |
| **Ethnicity** | | | 0.7715 |
| Not Hispanic or Latino | 29 (94) | 50 (100) | |
| Hispanic or Latino | 2 (6) | 0 (0) | |
| **Race** | | | 0.7566 |
| White | 31 (100) | 50 (100) | |
| Black | 0 (0) | 0 (0) | |

heterologous as shown in the Supplementary Table S2, but we don't have enough power to compare differences in immune response to homologous vs. heterologous vaccines due to the low number of heterologous boosted participants in the study. None of the participants reported SARS-CoV-2 breakthrough infection following vaccination. Since most of the participants received the BNT162b2 vaccine we did not segregate the antibody responses between the vaccine types. Therefore, all data analyses were conducted irrespective of the specific vaccine administered. Blood samples were collected 1 month after the second dose as well as 1 month and 4 months after the third vaccine dose (Fig. 1a, b).

Virus neutralizing titers were measured using pseudovirus neutralization assay (PsVNA) against the SARS-CoV-2 vaccine homologous WA1, as well as Omicron BA.1, BA.2 and BA.3 subvariants that were previously shown to correlate well with neutralization titers measured with authentic SARS-CoV-2 in plaque reduction neutralization tests[4,5]. A PsVNA50 titer of 1:60 was used as a seropositive cut-off based on current understanding of neutralizing antibody as correlate of protection against COVID-19[6].

Ninety-four percent (94%) of previously naïve individuals had positive WA1-neutralizing antibody titers (>1:60) at 1 month after the second vaccination with geometric mean titers (GMT) of 396. Following a third vaccine dose, 100% of naïve adults were seropositive with 5.9-fold average increase in neutralization titers (GMT = 2333) compared with the second vaccination. However, by four months after the third vaccine dose, the neutralization titers were 3.2-fold lower (GMT = 729) with 96% naïve individuals with PsVNA50 > 1:60 (Fig. 1c). In comparison with the naïve group, 97% of SARS-CoV-2 convalescent individuals were seropositive after two vaccinations with higher PsVNA50 titers (GMT = 1680) against vaccine-homologous WA1. After a third vaccination, WA1 neutralization titers increased by 2-fold (GMT = 3501) with 100% seropositive. Importantly, the convalescent cohort maintained high titers after 4 months post-third vaccine dose with only 1.7-fold drop in neutralization titers (GMT = 2024) and 100% seropositivity (Fig. 1c). These findings confirmed earlier reports on the superiority of hybrid immunity, and further demonstrate the longevity of higher vaccine homologous WA1 neutralizing titers compared with vaccinated naïve individuals.

Neutralization titers were significantly reduced against all three Omicron subvariants compared with WA1 (Fig. 1d). After two vaccinations of naïve individuals only 12–16% had seropositive neutralization titers >1:60 against BA.1, BA.2, and BA.3 with 22-25-fold reduction in PsVNA50 titers compared with WA1. However, after a third vaccine dose, 88–90% individuals demonstrated Omicron cross-neutralizing antibodies with PsVNA50 GMTs ranging between 243-312 (only 6–10-fold lower neutralization titers than WA1) (Fig. 1d). In comparison, the convalescent cohort exhibited higher responses against the Omicron subvariants BA.1, BA.2 and BA.3, after two vaccinations (56–71% seropositive, GMTs=87–146), that were further boosted after the third vaccine dose reaching 97%-100% seropositivity and GMTs ranging between 338 and 534 (7–10-fold lower than WA1 at 1-month post-third vaccine dose) (Fig. 1d).

### Durability of neutralizing antibodies following 3rd vaccination in naïve vs convalescent individuals

We also determined the persistence of neutralizing antibodies against Omicron subvariants at four months post 3rd vaccine dose (Fig. 1e and Supplementary Figure S1) in a subset of vaccinated individuals. The reduction in neutralization titers for the naïve group (GMT of 90-99) was 2.5–3.2-fold, while reduction in titers for the convalescent group (GMT of 194-233) was 1.6-2.2-fold compared with 1-month post-3rd vaccination time-point, similar to the reduction in neutralization titers against WA1 (Fig. 1c). Both the GMT and the percentage of individuals with neutralization titers above 1:60 against Omicron subvariants were higher for the convalescent group (88%) compared with the naïve

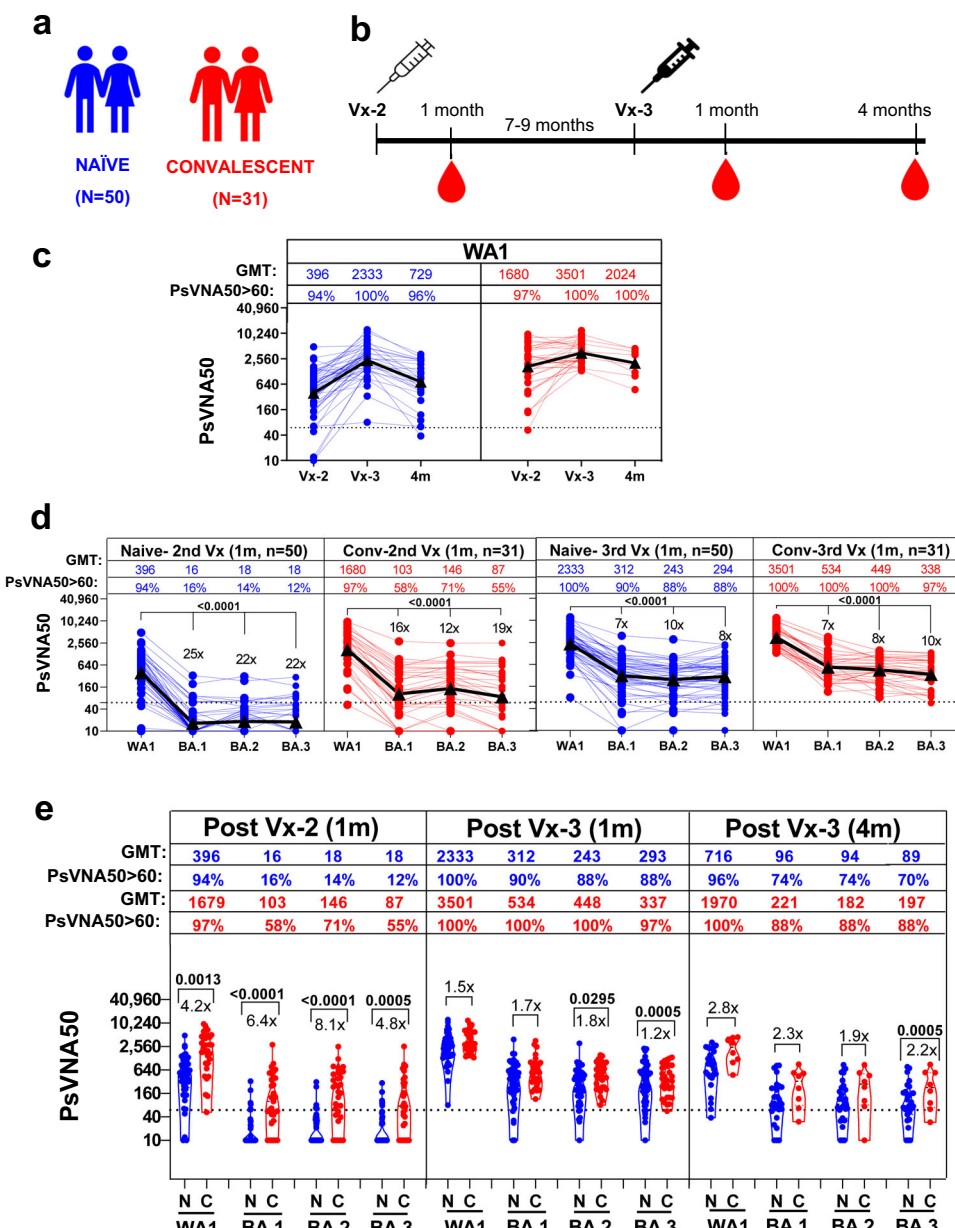

**Fig. 1 | Neutralizing antibodies following second and third vaccination in naïve and convalescent adults against SARS-CoV-2 WA-1 and Omicron subvariants.** **a** Overview of vaccination cohort, including 50 unexposed naive (N; in blue) and 31 COVID-19 convalescent (C; in red) adults receiving second and third mRNA vaccination. **b** Timeline of SARS-CoV-2 vaccination and sample collection in the two adult cohorts. **c–e** SARS-CoV-2 neutralizing antibody titers in naive (N; in blue) vaccination and convalescent adults (C; in red) one month after the second or third mRNA vaccination and four months post-3rd vaccination against SARS-CoV-2 WA1 and Omicron BA.1, BA.2 and BA.3. **c** PsVNA50 of post-2nd (Vx-2) or at 1 m and 4 m post-third dose (Vx-3) against WA1. Geometric mean PsVNA50 titers (GMT) are shown as black triangles and are presented for each vaccination time-point on top of the panel. Each data point represents an individual sample (circles). The horizontal dashed line indicates the seropositive cut-off for the neutralization titers (PsVNA50 of 60). Percent seropositivity (%S) for each group was calculated as number of seropositive samples divided by total number of samples x 100 in the group. The limit of detection for the neutralization assay is 1:20. Any sample that does not neutralize SARS-CoV-2 at 20-fold dilution was given a value of 10 for representation and data analysis purposes. **d** Plots showing PsVNA50 titers at 1 m after second or third mRNA vaccination serum of naive (N; in blue) and convalescent (C; in red) adults against WA1 and Omicron BA.1, BA.2 and BA.3. The fold-reduction in titers to Omicron BA.1 or BA.2 or BA.3 compared with WA1 are shown. **e** Comparisons of PsVNA50 titers against WA1 and Omicron BA.1, BA.2 and BA.3 for 1 m post-second or post-third mRNA vaccination or 4 m post-3rd vaccination from 50 naive (N; in blue) vs. 31 convalescent (C; in red) adults. Percent seropositivity is color coded for each of the group. The fold-difference in titers between naive vs. convalescent cohorts are shown. Differences between naive vs. convalescent cohorts were analyzed using Tukey's pairwise multiple comparison test that controlled for age, sex and BMI as covariates and the two-sided statistically significant p-values are shown. Nonsignificant *p* values (*p* > 0.05) are not shown. The data shown are average values of two experimental runs.

group (70–74%) at 4-months post-third vaccination (Supplementary Fig. S1 and Fig. 1e). It is important to note that after the 3rd vaccine dose, the neutralizing antibody responses against the vaccine homologous WA1 were not statistically different between the naïve vs. convalescent groups, while the difference between the responses to the highly transmissible Omicron subvariant BA.2 was statistically lower at 1 month (*p* = 0.0295) but not at 4 months after the third vaccine dose (Fig. 1e). The neutralization titers against Omicron BA.3 were statistically higher for the convalescent group compared with the naïve group at 1 month post-second vaccination (*p* = 0.0005), and at 1 month (*p* = 0.0005) as well as 4 months (*p* = 0.0005) after the third vaccine dose (Fig. 1e).

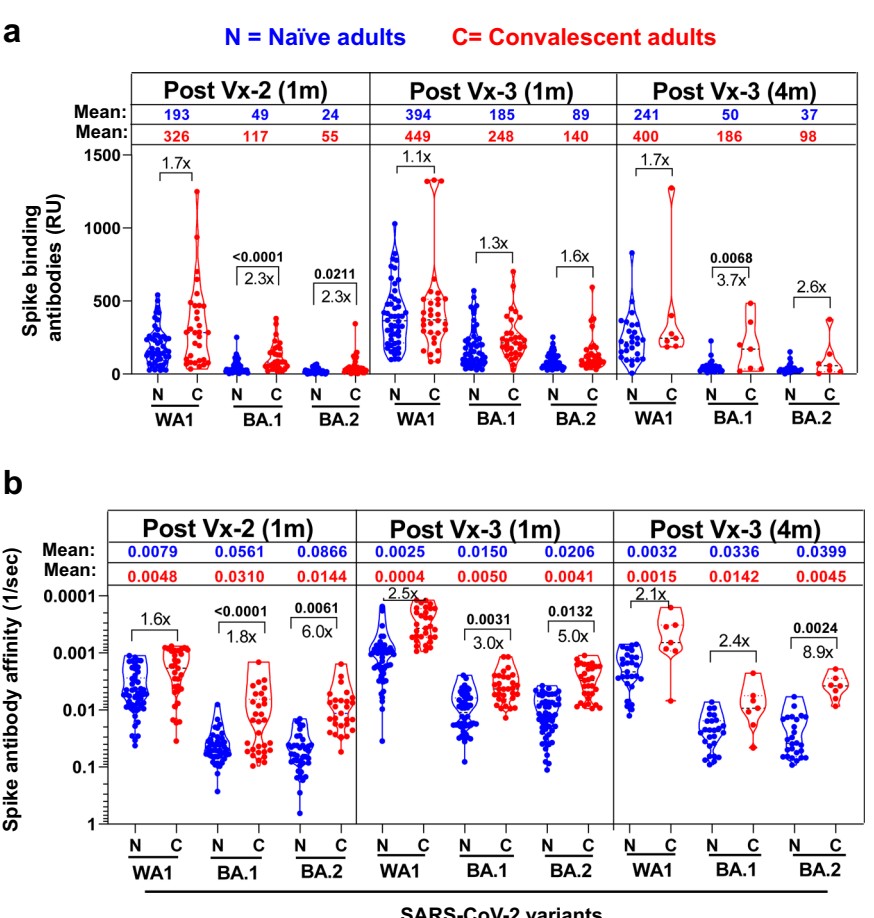

**Fig. 2 | Antibody affinity maturation of human antibody response to SARS-CoV-2 spike of WA1 and Omicron subvariants BA.1 or BA.2 following second and third SARS-CoV-2 mRNA vaccination in naïve vs. convalescent adults. a** Mean values ± range of total antibody binding (determined by maximum resonance units, Max RU) of 1:10 diluted serum samples of 50 unexposed naïve (N; in blue) adults vs. 31 COVID-19 convalescent (C; in red) adults at 1 m after second (Vx-2) or third mRNA vaccination (Vx-3) or 4 m post-3rd vaccination to purified trimeric spike of WA1 or Omicron subvariants BA.1 or BA.2 by SPR. The mean values for Max RU for each spike are color coded by each group. **b** Polyclonal antibody affinity maturation (as measured by dissociation off-rate per sec) to SARS-CoV-2 spike proteins for 1 m after second (Vx-2) or third mRNA vaccination (Vx-3) or 4 m post-3rd vaccination for serum samples from 50 unexposed naïve (N; in blue) and 31 COVID-19 convalescent (C; in red), adults were determined by SPR. Antibody off-rate constants that describe the fraction of antibody-antigen complexes decaying per second were determined directly from the serially diluted post-vaccination sample interaction with SARS-CoV-2 spike proteins using SPR in the dissociation phase as described in Materials and Methods. Off-rate was calculated and shown only for the sample time points that demonstrated a measurable (>10RU) antibody binding in SPR. Antibody affinity was not determined for those serum whose spike-binding antibodies were <10RU. The mean values are color coded by each group. The fold-difference in antibodies between naïve vs convalescent cohorts are shown. All SPR experiments were performed twice and the researchers performing the assay were blinded to sample identity. The variation for each sample in duplicate SPR runs was <6%. The data shown are the average value of two experimental runs. Differences between naïve vs. convalescent cohorts were analyzed by lme4 and emmeans packages in R using Tukey's pairwise multiple comparison test that controlled for age, sex and BMI as covariates and the two-sided statistically significant p-values are shown. Nonsignificant p values (p > 0.05) are not shown.

To identify potential role of sex on vaccination-induced immune response, we compared the neutralizing antibody response of males' vs females after second and third mRNA vaccination (Supplementary Fig. S2). The PsVNA50 titers against WA1 and the Omicron variants were not significantly different between males vs. females in either convalescent group or in the naïve group. However, at 4 months post-third vaccine dose, the convalescent males exhibited higher neutralization titers than convalescent females against WA.1 (1.8-fold), BA.1 (2.4-fold), BA.2 (2.7-fold) and BA.3 (2.5-fold) (Supplementary Fig. S2).

The neutralizing antibody titers following second mRNA vaccination in convalescent individuals correlated significantly with the time interval between infection and first vaccination suggesting a longer time interval between infection and first-vaccination results in higher SARS-COV-2 neutralization antibody response induced by vaccination (Supplementary Fig. S3). No correlation was observed for post-third vaccination neutralization titers and infection-vaccination

time-interval, suggesting that with time, the antibody responses even in individuals with hybrid immunity reaches a plateau.

These findings demonstrated that hybrid immunity following infections with early SARS-CoV-2 strains followed by mRNA vaccination with the ancestral strain provides superior neutralizing antibody response with significantly higher cross-neutralization of Omicron subvariants BA.1, BA.2 and BA.3, either after two or three doses of mRNA vaccination.

## Vaccination-induced binding antibodies against spike trimeric proteins of SARS-CoV-2 WA1 and Omicron BA.1 and BA.2 variants

Surface Plasmon Resonance (SPR) was used to measure total antibody binding against recombinant spike proteins derived from vaccine homologous WA1 and currently predominant circulating Omicron BA.1 and BA.2 subvariants as determined by Resonance Units (RU) (Fig. 2a and Supplementary Fig. S4)[7]. The purified recombinant SARS-CoV-2

spike proteins were captured to a Ni-NTA sensor chip with low protein density (200 RU) on the chip such as to measure monovalent interactions independent of the antibody isotype, as described before[5,8–11].

Antibody binding to the BA.1 and BA.2 spike proteins were lower than binding of the vaccine homologous WA1 spike after two or three mRNA vaccinations for both cohorts (Fig. 2a and Supplementary Fig. S4 and S5). The differences in spike-binding antibodies between the naïve vs. hybrid immunity groups were statistically significant after the second vaccine dose against Omicron BA.1 ($p < 0.0001$) and BA.2 ($p = 0.0211$) spikes, and against BA.1 spike at 4 months after the third vaccine dose ($p = 0.0068$) (Fig. 2a).

### Antibody affinity of WA1 and Omicron spike-binding antibodies following SARS-CoV-2 mRNA vaccination in naïve vs convalescent individuals

To determine qualitative difference in antibody avidity maturation against SARS-CoV-2 spike proteins induced following mRNA vaccination, antibody dissociation kinetics (off-rate constants), which describe the stability of the antigen-antibody complex, i.e., the fraction of complexes that decay per second in the dissociation phase, that are independent of antibody concentration were used as a surrogate for overall average avidity of polyclonal antibody were determined directly from the human polyclonal sample interaction with recombinant purified SARS-CoV-2 spike proteins of vaccine homologous WA1 as well as Omicron BA.1 and BA.2 using SPR in the dissociation phase only for the sensorgrams with Max RU in the range of 10–150 RU, as described before[5,7–14]. Previously, we showed that antibody kinetics measured under optimized SPR conditions primarily represent the monovalent interactions between the antibody-antigen complex, as antigen-antibody binding off-rates of the IgG and Fab interaction with protein antigens were similar[5,8,10,14].

Antibody binding dissociation rates against WA1 spike were slower indicating higher binding affinity after the third vaccination compared with the second vaccination against the vaccine spike protein (Fig. 2b and Supplementary Fig. S6). The antibody binding affinity against WA1 in the hybrid immunity group trended higher than the naïve group at one month after the 2nd vaccine dose (1.6-fold), one month after the 3rd dose (2.5-fold) and four months after the 3rd dose (2.1-fold) but these differences did not reach statistical significance (Fig. 2b).

Antibody binding affinity against the spike protein of Omicron subvariants BA.1 and BA.2 trended lower compared with the WA1 spike, but an increase in antibody affinity (i.e., slower dissociation rates) was observed one month after the 3rd dose compared with post-2nd vaccine dose that was maintained at four months post-3rd vaccination (Supplementary Figure S6). Interestingly, binding antibody affinity were significantly higher for the hybrid immunity group compared with the naïve group against BA.1 and BA.2 spike proteins after the 2nd dose ($p < 0.0001$ and $p = 0.0061$, respectively), one month after the 3rd dose ($p = 0.0031$ and $p = 0.0132$, respectively) and at four months after the 3rd dose ($p = 0.0024$ for BA.2 spike) (Fig. 2b and Supplementary Figs. S6 and S7).

The antibody evolution after mRNA vaccinations and their decay followed parallel trajectories with good correlation between PsVNA50 neutralizing antibody titers and spike antibody binding (Supplementary Fig. S8) as well as antibody affinity (Fig. 3). Importantly, individuals with hybrid immunity demonstrated durable higher neutralizing antibody titers and higher antibody affinity than naïve adults at least up to 4 months post-third vaccination against the Omicron subvariants, especially BA.2 (Fig. 4), which recently has become dominant around the globe, and continues to evolve further.

## Discussion

Recent infections with circulating SARS-CoV-2 Omicron subvariants (especially BA.1 and BA.2) are determined by multiple factors including immune escape due to specific amino acid mutations in the spike, the levels (specificity and affinity) of cross-reactive polyclonal antibodies elicited by prior infection and/or vaccination, and waning immunity[15–17].

Our study demonstrates that a third vaccination significantly boosts neutralizing antibodies against the Omicron subvariants, including BA.1 and BA.2, as recently reported[18], as well as BA.3. But in our study, convalescent individuals with hybrid immunity showed better antibody response against the rapidly spreading Omicron BA.1, BA.2 and BA.3 compared with vaccinated and boosted individuals with no history of prior infection. Our data and other studies suggest that Omicron BA.3 may evade immunity acquired from vaccination slightly more efficiently than BA.1 and BA.2[19]. Therefore, the need for an additional booster vaccination may vary based on the individual's infection/vaccination histories and the circulating strains, in addition to various risk factors. In addition, our study demonstrated a strong correlation between SARS-CoV-2 neutralization titers and spike-binding antibody affinity against Omicron subvariants BA.1 and BA.2 spikes. Importantly, even at four months after the third dose the antibody binding affinities against the Omicron subvariants BA.1 and BA.2 were maintained and trended higher for the individuals with hybrid immunity. Recently, Wratil et al., reported similar findings but our data expands the antibody analyses, including delineating antibody affinity maturation and persistence of protective immunity to the Omicron subvariants up to 4 months post-third SARS-CoV-2 mRNA vaccination[20]. Although hybrid immunity due to prior SARS-CoV-2 infection followed by vaccination leads to broader antibody responses, it is important to understand that any infection still comes with a risk of complications. The gradual drop in antibody titers following hybrid immunity, irrespective if the infection took place before or after vaccination, suggest the immunity wanes at similar rates following vaccination and breakthrough infections with eventual loss of protection against circulating SARS-CoV-2 strains[21–23]. A possible role of immune imprinting in SARS-CoV-2 immune response due to prior SARS-CoV-2 infection/vaccination or the original antigenic sin (OAS) hypothesis, whereby adults with B-cell memory due to prior exposure to seasonal coronaviruses[8,13,14,24] requires further investigation. Recently, we observed anti-S2 cross-reactivity in naive older children but not in the younger children (<4 years old), who share homology with HKU1, 229E and OC43[13,25]. OAS was also observed in mice immunization studies with seasonal CoV followed by SARS-CoV-2 spike[26]. Most of these cross-reactive antibodies do not neutralize SARS-CoV-2 and do not contribute to SARS-CoV-2 neutralization.

Our data along with previous studies suggest the presence of circulating memory B cells with expanded affinity-matured repertoires in convalescent individuals that are recalled by vaccination and likely to re-enter secondary germinal centers to undergo further somatic hypermutation (SHM) and antibody affinity maturation[27–32]. In naïve individuals, antibody affinity maturation post vaccination lags the convalescent group but may continue to evolve in the weeks following additional vaccine boosts. The breadth of variant recognition is likely to be impacted both by the antigenic distance between the infecting and vaccination strains, the rate of antibody affinity maturation through repeated GC seedings of memory B cells, and the time post-infection and last vaccination[18,33,34].

The protective efficacy by vaccine-induced antibodies against emerging variants may be impacted by both specific amino acid mutations in the spike and the affinity of the polyclonal antibodies against the SARS-CoV-2 variants. An association was observed between high titers of low-affinity antibodies against RBD with the disease severity of COVID-19 patients[35]. In previous studies, we had demonstrated a strong correlation between antibody affinity and protection from highly pathogenic avian influenza viruses[36,37] and a correlation with clinical benefit in patients infected with Zika virus[38], Ebola virus[10], influenza virus[39] and COVID-19[8,14]. Therefore, in addition to virus

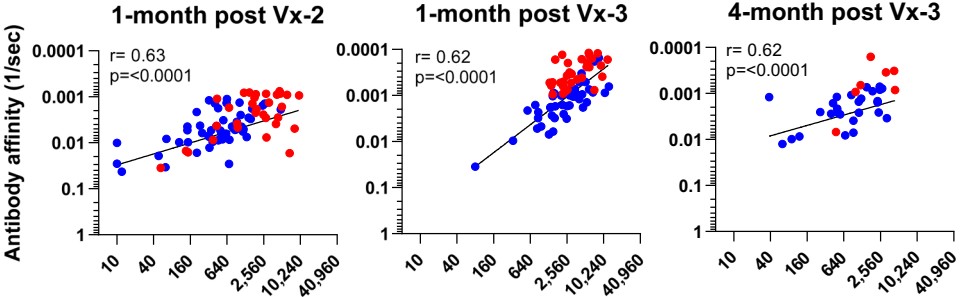

**a** Correlation between PsVNA50 titers and antibody affinity to WA1

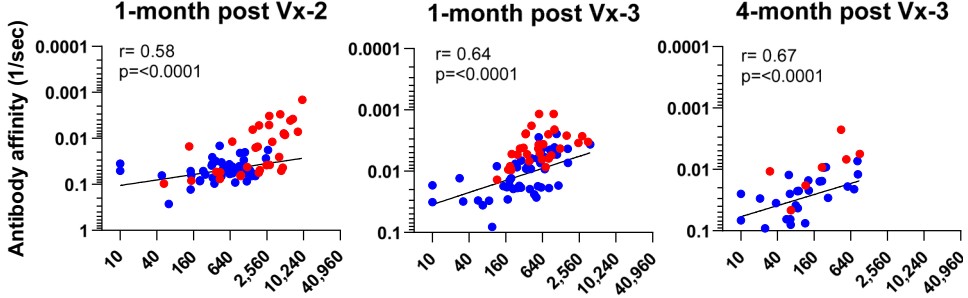

**b** Correlation between PsVNA50 titers and antibody affinity to BA.1

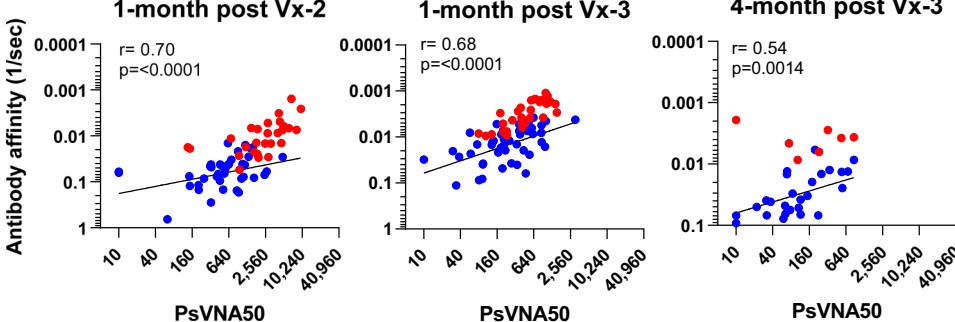

**c** Correlation between PsVNA50 titers and antibody affinity to BA.2

**N = Naïve adults** **C= Convalescent adults**

**Fig. 3 | Relationship of post-vaccination SARS-CoV-2 serum neutralizing antibodies in COVID convalescent and naïve adults with antibody affinity against SARS-CoV-2 spike protein.** Correlation analysis between serum PsVNA50 neutralization antibody titers generated following second and third vaccination of COVID exposed (*n* = 31) and unexposed naïve (*n* = 50) adults against vaccine-matched SARS-CoV-2 WA1 (**a**) and Omicron BA.1 (**b**) and BA.2 (**c**), and antibody affinity to spike protein of corresponding SARS-CoV-2 strain. Correlation analysis was performed using nonlinear regression model and associated Spearman's correlation coefficients (r) and regression significance (p) are shown for two-sided statistical test.

neutralization it is important to measure antibody affinity maturation against the SARS-CoV-2 spike not only for the vaccine strain, but also against spike proteins derived from the circulating variants of concern, that may influence the protective efficacy of vaccines against current and emerging SARS-CoV-2 variants of concern.

No breakthrough infections with Omicron were reported in our cohorts. However, it will be important to demonstrate the impact of high-affinity anti-BA.1/BA.2 antibodies not only on severe outcome (hospitalization/death) but also on the clinical course and viral loads in Omicron breakthrough infections. Based on the waning immunity, it is expected that naïve adults will potentially require another vaccine booster earlier (4–6 months post-third vaccination) than individuals with hybrid immunity to generate protective strong neutralizing antibodies especially against the rapidly spreading Omicron subvariants BA.2.12.1, BA.4 and BA.5. In summary, our study demonstrates that hybrid immunity provides superior immune kinetics, breadth, and durable high-affinity antibodies vs. vaccine-only immunity against Omicron sublineages up to four months post-third vaccination with mRNA vaccines.

## Methods

### Study background, patient characteristics, vaccines

Heat-inactivated de-identified samples were obtained from participants enrolled in the SPARTA (SARS2 Seroprevalence and Respiratory Tract Assessment) program in Athens, GA (USA) with written informed consent (Supplementary Table S1). The study procedures, informed

**Fig. 4 | Durability of vaccination-induced SARS-CoV-2 vaccination-induced immunity against Omicron BA.2.** Schematic highlighting the durability of SARS-CoV-2 vaccination-induced antibody response in naïve (blue) vs convalescent (red) adults shown as percent seropositivity of neutralization titers and spike antibody affinity against SARS-CoV-2 Omicron BA.2 following second and third mRNA vaccination.

consent, and data collection documents were reviewed and approved by the WIRB-Copernicus Group Institutional Review Board (WCG IRB #202029060) and the University of Georgia. All patients provided written informed consent. Consent on this study included agreement for the use of remnant material for additional immunological assays at the time of study enrollment. Samples were tested in different antibody assays with approval from the U.S. Food and Drug Administration's Research Involving Human Subjects Committee (FDA-RIHSC) under exemption protocol '252-Determination- CBER-2020-08-19. All samples were tested in duplicates in a blinded fashion. Most absolute values and fold-change graphs were normalized to Log2 for statistical calculations.

Table 1 lists the summary of patient characteristics for this cohort. Supplementary Table S2 lists the extended demographics and vaccine/booster types, and the time point of sampling relative to the COVID-19 vaccination. Sex and/or gender was considered in the study design and was determined based on self-report by the study participants. All participants reported to be binary (either Male or Female).

Post-SARS-CoV-2 mRNA vaccination serum samples were collected after the second vaccination and third vaccination from 31 convalescent individuals who were exposed to SARS-CoV-2 but not hospitalized (had confirmed SARS-CoV-2 infection between March and November 2020) prior to circulation of any of the Omicron lineages as well as naïve 50 healthy adults[40]. During March – November 2020, the predominant circulating SARS-CoV-2 strains in the US were the D614G strain and the Alpha variant prior to vaccines being available in the US. All immune naïve individuals tested negative by RT-PCR for nucleic acid and did not report any COVID-like symptoms at any point during the study. They also tested antibody negative to SARS-CoV-2 spike, prior to vaccination. All subjects received mRNA vaccine either from Moderna (mRNA-1273) or from Pfizer (BNT162b2) at 4-week or 3-week intervals between doses, respectively, between January and February 2021. All of them then received a third mostly homologous vaccine booster dose in September-October 2021. The primary series with the first two vaccine doses were homologous for each vaccinee. The boosters were either homologous or heterologous as shown in the Supplementary Table S2. None of the participants reported SARS-CoV-2 breakthrough infection following vaccination.

**Neutralization assay**
Sera were evaluated in a qualified SARS-CoV-2 pseudovirion neutralization assay (PsVNA) using SARS-CoV-2 WA-1 strain and the three Omicron variant lineages: BA.1 (B.1.1.529), BA.2 and BA.3 (Supplementary Table S1). SARS-CoV-2 neutralizing activity measured by PsVNA correlates with PRNT (plaque reduction neutralization test with authentic SARS-CoV-2 virus) in previous studies[8,12,41].

Pseudovirions were produced as previously described[41]. Briefly, human codon-optimized cDNA encoding SARS-CoV-2 spike

glycoprotein of the WA1/2020 and variants were synthesized by GenScript and cloned into eukaryotic cell expression vector pcDNA 3.1 between the *BamH*I and *Xho*I sites. Pseudovirions were produced by co-transfection Lenti-X 293 T cells with psPAX2(gag/pol), pTrip-luc lentiviral vector and pcDNA 3.1 SARS-CoV-2-spike-deltaC19, using Lipofectamine 3000. The supernatants were harvested at 48 h post transfection and filtered through 0.45 μm membranes and titrated using 293T-ACE2-TMPRSS2 cells (HEK 293 T cells that express ACE2 and TMPRSS2 proteins)[41].

Neutralization assays were performed as previously described[8,12]. For the neutralization assay, 50 μL of SARS-CoV-2 S pseudovirions (counting ~200,000 relative light units) were pre-incubated with an equal volume of medium containing serial dilutions (20-, 60-, 180-, 540-, 1620-, 4860-, 14,580- and 43,740-fold dilution at the final concentration) of heat-inactivated serum at room temperature for 1 h. Then 50 μL of virus-antibody mixtures were added to 293T-ACE2-TMPRSS2 cells ($10^4$ cells/50 μL)[41] in a 96-well plate. The input virus with all SARS-CoV-2 strains used in the current study were the same ($2 \times 10^5$ relative light units/50 μL/well). After a 3 h incubation, fresh medium was added to the wells. Cells were lysed 24 h later, and luciferase activity was measured using One-Glo luciferase assay system (Promega, Cat# E6130). Antibody data was collected in MS Excel version 16.57. The assay of each serum was performed in duplicate, and the 50% neutralization titer was calculated using Prism 9 (GraphPad Software). Controls included cells only, virus without any antibody and positive sera. The limit of detection for the neutralization assay is 1:20. Any sample that does not neutralize SARS-CoV-2 at 20-fold dilution was given a value of 10 for representation and data analysis purposes. Two independent biological replicate experiments were performed for each sample and variation in PsVNA50 titers was <9% between replicates.

**Antibody binding kinetics to SARS-CoV-2 spike proteins by Surface Plasmon Resonance**
Steady-state equilibrium binding of post-SARS-CoV-2 vaccinated human polyclonal sample was monitored at 25 °C using a ProteOn surface plasmon resonance (BioRad). The purified recombinant SARS-CoV-2 spike proteins were captured to a Ni-NTA sensor chip with 200 resonance units (RU) in the test flow channels. Serial dilutions (10-, 50- and 250-fold) of freshly prepared samples in BSA-PBST buffer (PBS pH 7.4 buffer with Tween-20 and BSA) were injected at a flow rate of 50 μL/min (120 s contact duration) for the association, and disassociation was performed over a 600-s interval. Responses from the protein surface were corrected for the response from a mock surface and for responses from a buffer-only injection. Total antibody binding was calculated with BioRad ProteOn manager software (version 3.1). All SPR experiments were performed twice. In these optimized SPR conditions, the variation for each sample in duplicate SPR runs was <6%. The maximum resonance units (Max RU) shown in figures is for 10-fold diluted sample.

Antibody off-rate constants, which describe the stability of the antigen-antibody complex (i.e. the fraction of complexes that decays per second in the dissociation phase) were determined directly from the interaction of human polyclonal samples with recombinant purified SARS-CoV-2 proteins using SPR in the dissociation phase only for the sensorgrams with Max RU in the range of 10–150 RU and calculated using the BioRad ProteOn manager software for the heterogeneous sample model as described before[7,10,14]. Off-rate constants were determined from two independent SPR runs.

**Quantification and statistical analysis**
Descriptive statistics were performed to determine the geometric mean titer values and were calculated using GraphPad. All experimental data to compare differences among groups were analyzed using lme4 and emmeans packages in R (RStudio version 1.1.463).

The demographic characteristics of these study participants are shown in Supplementary Table S2. Since age, sex, and the body mass index (BMI) can be biologically plausible confounders, data were analyzed for statistical significance between convalescent vs unexposed naïve groups to control for age, sex, and BMI as covariates (predictor variables) using a multivariate linear regression model. To ensure robustness of the results, absolute measurements were log2-transformed before performing the analysis. For comparisons between the vaccine cohorts (factor variable), pairwise comparisons were extracted using 'emmeans' and Tukey-adjusted p values were used for denoting significance to reduce Type 1 error due to multiple testing. The tests were two-sided tests. The differences were considered statistically significant with a 95% confidence interval when the $p$ value was less than 0.05.

Samples were allocated randomly to each test group and tested blindly (researcher was blinded to sample identity) to minimize selection bias or detection bias. There were no exclusion criteria. All samples and data were used for analysis and presented in the study.

### Reporting Summary
Further information on research design is available in the Nature Research Reporting Summary linked to this article.

## Data availability
All data are shown in the manuscript figures and supplementary information. The complete dataset for this study are provided in the Source Data file. Source data are provided with this paper.

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

## Acknowledgements

We thank Keith Peden and Basil Golding for their insightful review of the manuscript. We thank Carol Weiss for providing plasmid clones expressing SARS-CoV-2 variants. The authors would like to thank all SPARTA program participants. The authors thank Hannah Hanley and Debbie Bratt for program coordination, as well as the member of the SPARTA collection, processing, and analysis teams, including Brittany Baker, Charlotte Bolle, Courtney Briggs, Jasmine Burris, Jasper Gattiker, Omar Hamwy, Lauren Howland, Hana Ji, Katie Mailloux, Cleopatria Smith, Terrie Waits, Emma Whitesell, and the entire staff at the University of Georgia Clinical and Translational Research Unit (CTRU). The antibody characterization work described in this manuscript was supported by FDA's MCMi grant #OCET 2021-1565 to S.K and intramural FDA-CBER COVID-19 supplemental funds. The SPARTA program was supported by the National Institute of Allergy and Infectious Diseases (NIAID), U.S. National Institutes of Health (NIH), Department of Health and Human Services contract 75N93019C00052, and the University of Georgia (US) grant UGA-001. T.M.R is also supported by the Georgia Research Alliance (US) grant GRA-001. The CTRU was supported by the National Center for Advancing Translational Sciences of the National Institutes of Health under Award Number UL1TR002378. The funders had no role in study design, data collection and analysis, interpretation, writing, decision to publish, or preparation of the manuscript. The content of this publication does not necessarily reflect the views or policies of the Department of Health and Human Services, nor does mention of trade names, commercial products, or organizations imply endorsement by the U.S. Government.

## Author contributions

All authors read and approved the final version of the manuscript. Designed research: S.K. and T.R. Performed research: L.B., G.G., F.Z. and S.K. Collected clinical samples and provided clinical data: D.F. and T.R. T.R. verified the underlying data. Contributed to Writing: H.G. and S.K. S.K. and H.G. verified the underlying data.

## Competing interests

The authors declare no competing interests.

## Ethics

The study at CBER, FDA, was conducted with de-identified samples and all assays performed fell within the permissible usages in the original consent. Antibody assays were performed with approval from the U.S. Food and Drug Administration's Research Involving Human Subjects Committee (FDA-RIHSC) under exemption protocol '252-Determination-CBER-2020-08-19.
