## [Peer Review File · Nature Communications]

Antibody affinity and cross-variant neutralization of SARS-CoV-2 Omicron BA.1, BA.2 and BA.3 following third mRNA vaccinationReviewers' Comments:

Reviewer #1:

Remarks to the Author:

Bellusci et al. report the durability of antibody affinity and cross-variant immunity against SARS-CoV-2 Omicron BA.1, BA.2, and BA.3 variants following 3rd mRNA vaccination in naïve versus convalescent individuals. The study is timely and important when the Omicron subvariants are causing new surges of COVID cases. The following points can substantiate the study.

1. Were the convalescent individuals most likely infected by the original SARS-CoV-2 (before the emergence of Alpha variant)? If so, please state this in the manuscript.
2. A reference should be added to justify "The antibody responses were not different between the vaccine types" (line 8).
3. Why 1:60 was defined as the cut-off line? This cut-off line is very different from the conventional plaque reduction neutralization assay. If 1:60 was the cut-off line, how the data points below 1:60 were plotted in the figures?

Reviewer #2:

Remarks to the Author:

The manuscript has investigated neutralizing responses against wild type SARS-CoV-2, Omicron BA.1, BA.2 and BA.3 response in vaccinated vs convalescent individuals after 2 and 3 doses. Although the neutralization is done with a pseudovirus assay and not with full virus, the study is well done and carefully assessed. In addition, antibody affinity was investigated and correlated with neutralization. It shows a well-described phenomenon that convalescent individuals mount a better response after vaccination than previously naïve individuals, although not much data were available so far for BA.2. The study shows in line with previous findings that a third vaccination improves immune response towards Omicron. Thus the study adds to the currently becoming more complex landscape of immunity background and further diversification of Omicron subvariants. The study is well done, informative and I have only minor comments.

Introduction: it is not clear here which constellation is exactly meant with hybrid immunity/convalescent, is it infection before vaccination or vaccine break through, please specify further. It is stated in the methods but for convenience of the reader, should be more precise in the abstract/introduction

Methods: How was the status of the immune naïve individuals determined? Was there an additional check for antibodies before vaccination, just to rule out unrecognized infections? How did the authors make sure there were no breakthrough infections in the individuals during the study?

It should be further specified if recipients received a full vaccination course with the same vaccine or if vaccines were combined. It is visible in the table but for convince of the reader should be mentioned in the text. Was there any difference seen when individuals were vaccinated with the same vaccine or had a mixed approach?

In the table, the time interval between infection and vaccination was rather variable between individuals – can the authors draw conclusions how a long vs a short time interval between infection and vaccination influences the neutralization response?

Results

Lines 190-191 Thus should be in the discussion not results: "Therefore, the need for an additional booster vaccination may vary based on the individual's infection/vaccination histories in addition to various risk factors." In general, I think the authors should be careful to draw conclusions on

vaccination schedules or recommendations based on neutralization data alone, as it is also important to see the real life efficacy towards hospitalization and death. It is also something that should be discussed that neutralizing data are not the full story and other factors (T cell, mucosal immunity) also contribute to protection.

General: Although hybrid immunity lead to broader antibody responses, it should be discussed somewhere that any infection still comes with a risk of complications, although in this study, infections took place in 2020, before a vaccine was available. It would also be interesting to discuss if there may be differences between hybrid immunity whether the infection took place before vaccination or after. There is currently a debate about immune imprinting, which should be mentioned in the discussion.

The authors could also discuss the relevance of adding BA.3 and how the three subvariants differed between each other

Food and Drug Administration
Center for Biologics Evaluation and Research
Office of Vaccines Research and Review
Division of Viral Products

RESPONSE TO REVIEWER'S COMMENTS:

Reviewer #1 (Remarks to the Author):

Bellusci et al. report the durability of antibody affinity and cross-variant immunity against SARS-CoV-2 Omicron BA.1, BA.2, and BA.3 variants following 3rd mRNA vaccination in naïve versus convalescent individuals. The study is timely and important when the Omicron subvariants are causing new surges of COVID cases.

Response: We thank the reviewer for the positive response and appreciating our study.

The following points can substantiate the study.

1. Were the convalescent individuals most likely infected by the original SARS-CoV-2 (before the emergence of Alpha variant)? If so, please state this in the manuscript.

Response: The convalescent adults were infected with SARS-CoV-2 during March – November 2020. At that time, the predominant circulating SARS-CoV-2 strains in the US were the D614G strain and the Alpha variant. We have added this information to both results and methods section.

Results Lines 81-82: During that time, the predominant circulating strain in the US were the D614G strain and the Alpha variant prior to availability of mRNA vaccines.

Methods Lines 311-314: During March – November 2020, the predominant circulating SARS-CoV-2 strains in the US were the D614G strain and the Alpha variant prior to vaccines being available in the US.

2. A reference should be added to justify “The antibody responses were not different between the vaccine types” (line 8).

Response: Since most of the participants received the BNT162b2 vaccine we did not segregate the antibody responses between the vaccine types. We have clarified this further in the revised manuscript.

Lines 89-91: Since the majority of the participants received the BNT162b2 vaccine we did not segregate the antibody responses between the vaccine types.

3. Why 1:60 was defined as the cut-off line? This cut-off line is very different from the conventional plaque reduction neutralization assay. If 1:60 was the cut-off line, how the data points below 1:60 were plotted in the figures?

Response: A PsVNA50 titer of 1:60 was used as a seropositive cut-off based on current understanding of neutralizing antibody as correlate of protection against

COVID-19 (Ref. 6). The virus neutralizing titers against SARS-CoV-2 and variants measured using our PsVNA were previously shown to correlate well with neutralization titers measured with authentic SARS-CoV-2 in plaque reduction neutralization tests (Reference 4, 5). The limit of detection for the PsVNA is 1:20. Any sample that does not neutralize SARS-CoV-2 at 20-fold dilution was given a value of 10 for representation and data analysis purposes.

We have added this information at various places in the revised manuscript.

Results Lines 94-99: Virus neutralizing titers were measured using pseudovirus neutralization assay (PsVNA) against the SARS-CoV-2 vaccine homologous WA1, as well as Omicron BA.1, BA.2 and BA.3 subvariants that were previously shown to correlate well with neutralization titers measured with authentic SARS-CoV-2 in plaque reduction neutralization tests (Ref 4, 5). A PsVNA50 titer of 1:60 was used as a seropositive cut-off based on current understanding of neutralizing antibody as correlate of protection against COVID-19 (Ref 6).

Methods Lines 340-342: The limit of detection for the neutralization assay is 1:20. Any sample that does not neutralize SARS-CoV-2 at 20-fold dilution was given a value of 10 for representation and data analysis purposes.

Figure legend Lines 609-611: The limit of detection for the neutralization assay is 1:20. Any sample that does not neutralize SARS-CoV-2 at 20-fold dilution was given a value of 10 for representation and data analysis purposes.

Reviewer #2 (Remarks to the Author):

The manuscript has investigated neutralizing responses against wild type SARS-CoV-2, Omicron BA.1, BA.2 and BA.3 response in vaccinated vs convalescent individuals after 2 and 3 doses. Although the neutralization is done with a pseudovirus assay and not with full virus, the study is well done and carefully assessed. In addition, antibody affinity was investigated and correlated with neutralization.

It shows a well-described phenomenon that convalescent individuals mount a better response after vaccination than previously naive individuals, although not much data were available so far for BA.2. The study shows in line with previous findings that a third vaccination improves immune response towards Omicron. Thus the study adds to the currently becoming more complex landscape of immunity background and further diversification of Omicron subvariants. The study is well done, informative and I have only minor comments.

Response: We thank the reviewer for appreciating our study and encouraging comments to make our manuscript better.

Introduction: it is not clear here which constellation is exactly meant with hybrid immunity/convalescent, is it infection before vaccination or vaccine break through, please

specify further. It is stated in the methods but for convenience of the reader, should be more precise in the abstract/introduction.

Response: The convalescent individuals first got SARS-CoV-2 infection in March-November 2020 and then got vaccinated and developed hybrid immunity. We have clarified this further throughout the manuscript.

Abstract Lines 37-40: The convalescent individuals who after SARS-CoV-2 infection got vaccinated developed hybrid immunity that showed broader neutralization activity and cross-reactive antibody affinity maturation against the Omicron BA.1 and BA.2 after either second or third vaccination compared with naïve individuals.

Intro Lines 65-67: In this study, we evaluated the capacity and durability of neutralizing antibodies and antibody affinity induced following mRNA-based (Pfizer-BioNTech BNT162b2 or Moderna mRNA-1273) vaccination in naïve versus convalescent individuals (infection before vaccination)....

Results Lines 74-76: In this study, we evaluated immune response following mRNA-based (Pfizer-BioNTech BNT162b2 or Moderna mRNA-1273) vaccination in a cohort of 81 adults: either naïve (N=50) or SARS-CoV-2 convalescent (infection before vaccination; N=31) individuals...

Lines 89-90: None of the participants reported SARS-CoV-2 breakthrough infection following vaccination.

Lines 159-162: These findings demonstrated that hybrid immunity following infections with early SARS-CoV-2 strains followed by mRNA vaccination with the ancestral strain provides superior neutralizing antibody response with significantly higher cross-neutralization of Omicron subvariants BA.1, BA.2 and BA.3, either after two or three doses of mRNA vaccination.

Methods: How was the status of the immune naïve individuals determined? Was there an additional check for antibodies before vaccination, just to rule out unrecognized infections? How did the authors make sure there were no breakthrough infections in the individuals during the study?

Response: All immune naïve individuals were SARS-CoV-2 PCR negative at baseline and have 1) tested Spike antibody negative prior to vaccination, 2) reported no COVID-like symptoms at any point during the study, and 3) did not test NAAT+ during any visit. No breakthrough infections were reported for the vaccinees described in the current manuscript.

We added the information in the revised manuscript.

Methods Lines 306-308: All immune naïve individuals tested negative by RT-PCR for nucleic acid and did not report any COVID-like symptoms at any point during the study. They also tested antibody negative to SARS-CoV-2 spike, prior to vaccination.

Lines 89-90: None of the participants reported SARS-CoV-2 breakthrough infection following vaccination.

Line 270: No breakthrough infections with Omicron were reported in our cohorts.

Lines 313-314: None of the participants reported SARS-CoV-2 breakthrough infection following vaccination.

It should be further specified if recipients received a full vaccination course with the same vaccine or if vaccines were combined. It is visible in the table but for convince of the reader should be mentioned in the text. Was there any difference seen when individuals were vaccinated with the same vaccine or had a mixed approach?

Response: We clarified it further in the revised manuscript.

Results Lines 86-89: The first two vaccine doses were homologous for each participant. The boosters were either homologous or heterologous as shown in the Supplementary Table S2, but we don't have enough power to compare differences in immune response to homologous vs. heterologous vaccines due to the low number of heterologous boosted participants in the study.

Methods Lines 311-313: The primary series with the first two vaccine doses were homologous for each vaccinee. The boosters were either homologous or heterologous as shown in the Supplementary Table S2.

In the table, the time interval between infection and vaccination was rather variable between individuals – can the authors draw conclusions how a long vs a short time interval between infection and vaccination influences the neutralization response?

Response: We performed correlation analysis and have included a new supplementary figure S3 and addressed it in the results section.

Lines 152-158: The neutralizing antibody titers following second mRNA vaccination in convalescent individuals correlated significantly with the time interval between infection and first vaccination suggesting a longer time interval between infection and first-vaccination results in higher SARS-COV-2 neutralization antibody response induced by vaccination (Supplementary Figure S3). No correlation was observed for post-third vaccination neutralization titers and infection-vaccination time-interval, suggesting that with time, the antibody responses even in individuals with hybrid immunity reaches a plateau.

Results

Lines 190-191 Thus should be in the discussion not results: “Therefore, the need for an additional booster vaccination may vary based on the individual’s infection/vaccination histories in addition to various risk factors.” In general, I think the authors should be careful to draw conclusions on vaccination schedules or recommendations based on neutralization data alone, as it is also important to see the real life efficacy towards hospitalization and death. It is also something that should be discussed that neutralizing data are not the full story and other factors (T cell, mucosal immunity) also contribute to protection.

Response: We moved the sentence to discussion section and further discussed the importance of vaccine efficacy against COVID-19.

Lines 227-229: Therefore, the need for an additional booster vaccination may vary based on the individual's infection/vaccination histories and the circulating strains, in addition to various risk factors.

Lines 259-269: The protective efficacy by vaccine induced antibodies against emerging variants may be impacted by both specific amino acid mutations in the spike and the affinity of the polyclonal antibodies against the SARS-CoV-2 variants. An association was observed between high titers of low affinity antibodies against RBD with disease severity of COVID-19 patients³⁵. In previous studies, we had demonstrated a strong correlation between antibody affinity and protection from highly pathogenic avian influenza viruses^{36,37} and a correlation with clinical benefit in patients infected with Zika virus³⁸, Ebola virus¹⁰, influenza virus³⁹ and COVID-19^{8, 14}. Therefore, in addition to virus neutralization it is important to measure antibody affinity maturation against the SARS-CoV-2 spike not only for the vaccine strain, but also against spike proteins derived from the circulating variants of concern, that may influence the protective efficacy of vaccines against current and emerging SARS-CoV-2 variants of concern.

General: Although hybrid immunity lead to broader antibody responses, it should be discussed somewhere that any infection still comes with a risk of complications, although in this study, infections took place in 2020, before a vaccine was available. It would also be interesting to discuss if there may be differences between hybrid immunity whether the infection took place before vaccination or after. There is currently a debate about immune imprinting, which should be mentioned in the discussion.

Response: This is an important point and we have discussed in the discussion section.

Lines 236-249: Although hybrid immunity due to prior SARS-CoV-2 infection followed by vaccination leads to broader antibody responses, it is important to understand that any infection still comes with a risk of complications. The gradual drop in antibody titers following hybrid immunity, irrespective if the infection took place before or after vaccination, suggest the immunity wanes at similar rates following vaccination and breakthrough infections with eventual loss of protection against circulating SARS-CoV-2 strains^{21, 22, 23}. A possible role of immune imprinting in SARS-CoV-2 immune response due to prior SARS-CoV-2 infection/vaccination or the original antigenic sin (OAS) hypothesis, whereby adults with B-cell memory due to prior exposure to seasonal coronaviruses^{8, 13, 14, 24} requires further investigation. Recently, we observed anti-S2 cross-reactivity in naive older children but not in the younger children (<4 years old), who share homology with HKU1, 229E and OC43^{13, 25}. OAS was also observed in mice immunization studies with seasonal CoV followed by SARS-CoV-2 spike²⁶. Most of these cross-reactive antibodies do not neutralize SARS-CoV-2 and do not contribute to SARS-CoV-2 neutralization.

The authors could also discuss the relevance of adding BA.3 and how the three

subvariants differed between each other

Response: We have addressed it in the revised discussion section.

Lines 221-227: Our study demonstrates that a third vaccination significantly boosts neutralizing antibodies against the Omicron subvariants including BA.1 and BA.2, as recently reported¹⁸, as well as BA.3. But in our study, convalescent individuals with hybrid immunity showed better antibody response against the rapidly spreading Omicron BA.1, BA.2 and BA.3 compared with vaccinated and boosted individuals with no history of prior infection. Our data and other studies suggest that Omicron BA.3 may evade immunity acquired from vaccination slightly more efficiently than BA.1 and BA.2 ¹⁹.